# Research Progress and Development Trends of Greenhouse Gas Emissions from Cereal–Legume Intercropping Systems

**Fanyun Yao** [1], **Yang Wu** [2], **Xiaodan Liu** [1], **Yujun Cao** [1], **Yanjie Lv** [1], **Wenwen Wei** [1], **Wenhua Xu** [1], **Zhiming Liu** [1], **Jie Liang** [1] and **Yongjun Wang** [1,*]

1   Institute of Agricultural Resources and Environment, Jilin Academy of Agriculture Sciences, Changchun 130033, China; yaofanyun@163.com (F.Y.); ymliuxiaodan@163.com (X.L.); caoyujun828@163.com (Y.C.); lvyanjie@cjaas.com (Y.L.); xingfu_0070623@163.com (W.W.); ecowhxu@126.com (W.X.); liuzhiming5050@163.com (Z.L.); liangjie9669@163.com (J.L.)
2   Institute of Jiangxi Oil-Tea Camellia, Jiujiang University, Jiujiang 332005, China; yangwu15@126.com
*   Correspondence: yjwang2008@cjaas.com; Tel.: +86-431-8706-3941; Fax: +86-431-8706-3165

**Abstract:** High yields and low carbon emissions are new challenges for modern crop production. Balancing the crop yield and reducing greenhouse gas (GHG) emissions has become a new field of agronomic technology innovation. Cereal–legume intercropping is a typical diversification planting system, which has been expected to achieve the dual goals of high production and low GHG emissions. However, the synergistic effect of integrating various technologies in an intercropping system on GHG emissions and whether it will achieve the high yield and low emissions goal remains to be determined. Therefore, bibliometric analysis has investigated the worldwide development trend of cereal–legume intercropping designs. The literature on the GHG emissions of the cereal–legume intercropping system was summarized. Additionally, the effects and mechanisms of different agricultural management methods regarding soil nitrous oxide and carbon dioxide emissions in the cereal–legume intercropping system were summarized. The research on GHG emissions of cereal–legume intercropping systems in non-growing seasons must be revised. In situ observations of GHG emissions from intercropping systems in different regions should be strengthened. This work is valuable in supporting and evaluating the potential of GHG reduction in a cereal–legume intercropping system in various farming areas.

**Keywords:** bibliometric analysis; soil greenhouse gas emissions; crop residue retention; nitrogen fertilizer; tillage

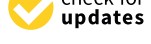



## 1. Introduction

Mitigating global warming caused by greenhouse gas (GHG) emissions is challenging for all of humanity. As emphasized by the International Panel on Climate Change (IPCC) Special Report on Global Warming of 1.5 °C (SR1.5) in 2018, drastic measures are urgently needed to reduce the risks and effects of climate change by limiting the increase in the global average temperature to 1.5 °C above the pre-industrial level [1]. Agricultural production has been considered one of the essential contributors to the anthropogenic source of non-carbon greenhouse gases (GHG$_S$) [2]. It is estimated that the food system was responsible for 25% to 30% of global emissions, around one-third if we include all agricultural products [3]. Meanwhile, agriculture also has considerable potential to mitigate climate change [4].

GHG emissions from farmland soil greatly depend on cultivation practices. Intercropping, as a classical multi-cropping system, has been widely demonstrated to enhance the crop yield and nutrient use efficiency by exploiting niche crop and seasonal differentiation and positive interactions between organisms, if appropriately managed. Therefore, intercropping is a crucial way to achieve the dual goals of increasing crop yields and reducing GHG emissions [5–8]. The primary consideration for intercropping is selecting compatible

crops to minimize competitive inhibition, allow for the ease of field management, and increase the profit per land unit compared to monoculture [9]. Cereal–legume intercropping is the most widely used combination globally and is an essential component of climate-smart agriculture and sustainable intensification in some regions [10]. However, intercropping will affect the soil's physical and chemical properties, influencing microbial-mediated GHG emissions [11]. Many studies have shown a reduction in soil carbon dioxide ($CO_2$) and $N_2O$ emissions in a cereal–legume system compared to monoculture [11–13]. However, the reason behind this still needs to be clarified, with limited systematic studies on GHG emissions with cereal–legume intercropping.

Bibliometrics has been widely used to study the frontier progress of many disciplines or fields since the 21st century. Its theories and methods have been extensively used to evaluate and predict the historical process, current situation, development trend, and hotspots of various research fields. Researchers have analyzed the research trends of global greenhouse gas emissions [14] and intercropping (including agroforest, crop mixture, vegetable legume, cereal legume, etc.) [15]. Still, a comprehensive and objective statistical report on the achievements must be provided. The objectives of the study were (1) to analyze the development trend of cereal–legume intercropping through a bibliometric method, (2) to summarize the emission characteristics and mechanism of soil $N_2O$ and $CO_2$ and elaborate the main factors affecting soil GHG emissions from intercropping systems, and (3) to propose three issues (compound effect of multi-technology, indirect GHG emissions and non-growing season GHG emissions) that should be strengthened in the study of GHG emissions from cereal–legume intercropping systems.

## 2. Analysis of the Development Trend of Cereal–Legume Intercropping

The article uses the information visualization software CiteSpace [16] to study the research related to cereal–legume intercropping between 2000 and 2021 in the Web of Science core collection (WoSCC) database. The query sets used for the literature search were TS = (legume OR soybean OR peanut OR pea OR bean) AND (interplant OR intercropping). Diversity in languages further restricted the search results and document types. Therefore, only articles written in English were retrieved. Two thousand and two hundred publications from 2005–2021 were obtained and saved as text files (see Supplementary materials for details) containing the "full record with citation data". In the final step, 839 relevant papers were selected by manual screening one by one (removing the literature about legume intercropping with noncereal crop plants and ensuring all works of literature were research articles). At the same time, the CiteSpace version. 5.6.5 was used to remove duplications.

### 2.1. Annual Trend in Publications and the Top Contributing Institutions

The trend of the number of publications and institutions in the database was analyzed (Figure 1). As shown in Figure 1, the number of publications in this field was relatively small and steady before 2016, with an average of 33 per year. After 2017, it increased rapidly, and the emphasis on cereal–legume intercropping increased; for example, in 2021, there were 137 publications. Figure 2 shows the Institution Collaboration Network in cereal–legume intercropping from 2005 to 2021. The thicker connection line indicates the closer cooperation between institutions, and each link between two different institutions is represented by a spectrum of colors corresponding to the years of occurrence. The 17 colors, from light gray to red, correspond to 2005 to 2021. The size of the nodes represents the number of papers published by the author's institution. The top five institutions were the China Agricultural University, Sichuan Agricultural University, the Ministry of Agriculture, the Chinese Academy of Agricultural Sciences, and Wageningen University in the Netherlands. Previous research results showed that the World Agroforestry Center ranked first in terms of publication volume in all intercropping fields, followed by the Chinese Academy of Science and INRA [15]. However, the distribution of research institutions in Figure 2 reflects the current high level of attention to cereal–legume intercropping research in China, especially by the China Agricultural University and Sichuan Agricultural University. However, the

cooperation between these institutions needs to be more cohesive. Further exchanges and collaboration between different research institutions are required.

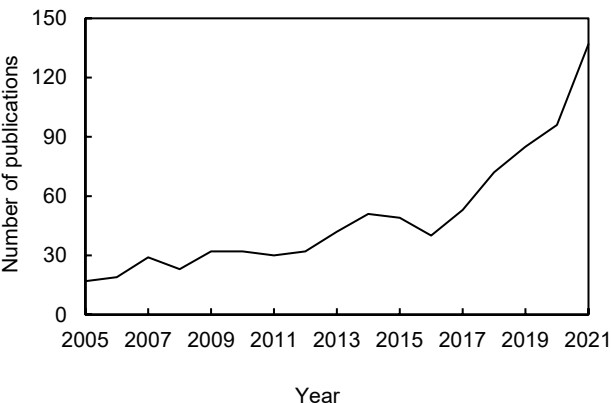

**Figure 1.** Annual trend in publications on research about cereal–legume intercropping based on data between 2005 and 2021 from WoSCC.

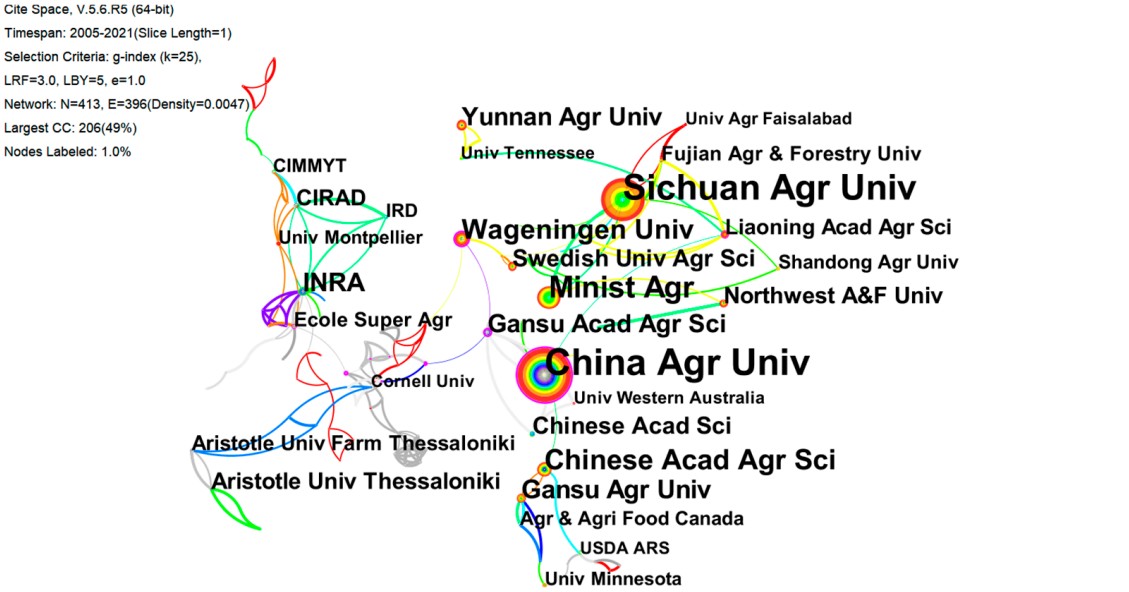

**Figure 2.** Distribution of research institutions in cereal–legume intercropping from 2005 to 2021.

### 2.2. Research Hotspots and Trend Analysis

The keywords in the paper reflect the main direction and core point of view of the research content. The keywords with the most substantial citations were used to illustrate the development and evolution of research directions [17]. We found that 93 keywords appeared more than 10 times through the data integration of CiteSpace software. The top 20 keywords with the most robust citation bursts between 2005 and 2021 are shown in Table 1. From the perspective of burst strength, the top five keywords were "cereal (S = 8.18)", "inorganic N (S = 6.21)", "tillage (S = 4.85)", "weed (S = 4.15)", and "light (S = 4.13)". This indicates that tillage practices, weed control, light resource utilization, and inorganic nitrogen dynamics in intercropping systems are receiving more attention. In terms of development time, "cereal", "weed", "density", "nitrogen fixation", "inorganic nitrogen", and "crude protein" appeared earlier, and these were the main areas of early research in the field of cereal–legume intercropping. "Greenhouse gas emissions" was a research hotspot between 2013 and 2015. "Weed" had the most prolonged duration and was the research hotspot between 2005 and 2015. "Intensification", "soybean", "light",

"crop productivity", and "climate change" are more recent burst words, which are still research hotspots from 2019 until now.

**Table 1.** Top 20 keywords with the most substantial citation bursts between 2005 and 2021.

| Keywords | Strength | Begin | End | 2005–2021 |
|----------|----------|-------|-----|-----------|
| cereal | 8.18 | 2005 | 2011 | |
| weed | 4.15 | 2005 | 2015 | |
| density | 3.97 | 2006 | 2008 | |
| nitrogen fixation | 2.79 | 2006 | 2010 | |
| inorganic nitrogen | 6.21 | 2007 | 2011 | |
| crude protein | 2.83 | 2007 | 2011 | |
| forage | 2.89 | 2008 | 2010 | |
| $N_2$ fixation | 3.60 | 2009 | 2010 | |
| nutrient acquisition | 3.21 | 2011 | 2014 | |
| nitrogen fertilization | 2.86 | 2011 | 2013 | |
| tillage | 4.85 | 2012 | 2015 | |
| greenhouse gas emission | 3.21 | 2013 | 2015 | |
| rhizosphere | 2.93 | 2014 | 2018 | |
| yield advantage | 3.50 | 2015 | 2019 | |
| interspecific interaction | 3.69 | 2017 | 2019 | |
| intensification | 2.58 | 2017 | 2021 | |
| soybean | 3.27 | 2018 | 2021 | |
| light | 4.13 | 2019 | 2021 | |
| crop productivity | 3.94 | 2019 | 2021 | |
| climate change | 3.13 | 2019 | 2021 | |

## 3. Effects of Intercropping on Soil Greenhouse Gas Emissions

### 3.1. Effects of Intercropping on Soil $N_2O$ Emissions

$N_2O$ emissions from the soil are an intermediate product in the nitrification–denitrification reaction [18,19]. Soil physicochemical properties and microbial community diversity are changed with increased crop diversification, resulting in changes in soil $N_2O$ emissions [20]. Differences in $N_2O$ emissions can also result from intercropping treatments that use different legume species and cultivars [21]. In contrast to conventional maize cultivation based on monocropping, the nitrogen fixation of legume crops and the transfer of nitrogen between maize and legume crops significantly influenced the nitrogen cycle in intercropping systems [22]. It was found that maize–peanut intercropping could reduce soil $N_2O$ emissions by 13% compared with maize monoculture [13]. Soil $N_2O$ emissions in maize–soybean intercropping were 32.0% and 47.8% lower than those in maize and soybean, respectively [23]. It is assumed that the soil water content was the main reason for the difference in soil $N_2O$ emissions in different intercropping systems. There was no significant difference in the $N_2O$ emission flux between monocultures and intercropping systems when the soil water content was <15% [24]. Under aerobic conditions, other environmental and soil physio-chemical factors might be more important than soil moisture. Excessive soil moisture may lead to loss of dissolved nitrogen, leading to increased indirect emissions of $N_2O$ [10]. Most studies showed that, under the same nitrogen application level, cereal–legume intercropping significantly reduced soil $N_2O$ emissions in the whole

growth period of crops compared with cereal monoculture [10–13,25]. This may be related to the improving nitrogen use efficiency in cereal–legume intercropping [11,26].

*3.2. Effects of Intercropping on Soil $CO_2$ Emissions*

The heterotrophic respiration of microorganisms and autotrophic respiration from roots are the primary sources of $CO_2$ emissions in soil. In addition to soil physicochemical properties, fertilization, irrigation, and the planting system also affect soil $CO_2$ emissions [27]. However, there are still many uncertainties about the effects of cereal–legume intercropping on soil $CO_2$ emissions. Dyer et al. [28] reported that, compared with maize monoculture and soybean monoculture, maize–soybean intercropping significantly reduced the total soil $CO_2$ emissions in the crop growing season by 18.1–20.4%. Ma et al. [13] concluded that maize–peanut intercropping reduced soil $CO_2$ emissions by 23.4%. However, there is still some variance in the results. It was found that there was no significant difference in soil $CO_2$ emissions between intercropping and monoculture treatments despite the cumulative $CO_2$ emissions in maize intercropping treatments being 4.0–8.9% lower than those in maize monoculture treatments during the whole growth period [29]. Sweet corn and soybean intercropping with crop row ratios of 2:3 and 2:4 increased the $CO_2$ emission compared with sweet corn monocropping. Additionally, $CO_2$ emissions were shown to increase with increased nitrogen application rates [30]. Rainfall variability, differing soil types across test sites, and variable sampling methods may have added to the inconsistency in the results [29].

## 4. Effects of Diversified Intercropping on GHG Emissions

Based on the diversification and refinement of basic research on cereal–legume intercropping systems in recent years, we summarized the effects of different management measures such as the intercrop row ratio and in-crop strip width, crop residue retention, and no-tillage on GHG ($CO_2$ and $N_2O$) emissions from intercropping systems. Methane emissions were not considered, as emissions from rain-fed farming systems are usually negligible, and few studies exist.

*4.1. Effects of Spatial Arrangements on GHG Emissions in Intercropping Systems*

Crop yields in the intercropping systems are closely related to spatial arrangements (mainly border–row proportion, bandwidth (crop rows per strip), and planting density) [31–33]. The field configuration can influence the efficiency of resource utilization and GHG emissions. Wang et al. [30] reported that both the GHG emissions intensity and global warming potential of sweet corn and soybean intercropping with different crop row ratios did not differ significantly. Dyer et al. [28] observed that different intercropping ratios (1:2 and 2:3) did not substantially affect soil $CO_2$ and $N_2O$ emissions. These results were confirmed by other studies on sweet corn–soybean intercropping [30,34]. In addition, the 2:2 equal row spacing and narrow row width ratio had no significant effect on soil GHG emissions [28]. According to the literature we have collected [28–30,34], the border–row proportion and bandwidth do not significantly affect the GHG emissions of a cereal–legume intercropping system.

Plant density is one of the leading management practices to affect the yield in cropping systems [33], especially for intercropping [35]. Therefore, producers often improve the intercropping yield by increasing the plant density of component crops [36]. Luo et al. [37] showed that plant density slightly impacted $N_2O$ emissions. However, the result obtained by Yang [38] through a three-year field experiment showed that increasing the maize density increased the carbon emissions. Other studies on plant density's effect on GHG emissions under the intercropping system exist. However, whether increasing plant density can reduce soil GHG emissions while ensuring viable intercropping production is still being determined. Further investigations are required to clarify its emission reduction potential and mechanisms.

### 4.2. Effects of Nitrogen Fertilizer Management on GHG Emissions from Intercropping Systems

Optimized nitrogen fertilizer management is essential for crop yields and carbon sequestration [4,30]. Nitrogen fertilizer application significantly contributes to $N_2O$ emissions from agriculture, and mitigating soil $N_2O$ emissions is vital for staying below a 1.5 °C warming threshold [39]. Studies have shown that intercropping with legume crops and an appropriate pre-plant or basal nitrogen fertilizer split application could significantly mitigate GHG emissions while ensuring the crop yield. Tang et al. [40] concluded that, under two rows of sweet corn and four rows of soybean intercropping patterns, a reduction in nitrogen fertilizer application could significantly reduce the soil mean cumulative $N_2O$ emission in six seasons but had no significant effect under an intercropping pattern of two rows of sweet corn and three rows of soybean. Fu et al. [41], through a long-term location experiment, showed that maize–soybean intercropping with reduced nitrogen application (180 kg/hm$^2$) could reduce GHG emissions compared to traditional nitrogen application (240 kg/hm$^2$). However, a recent study concluded that the sweet corn–soybean 2:4 intercropping patterns with 300 kg N ha$^{-1}$ significantly affected crop yields and reduced net GHG emissions compared to conventional sweet corn farming systems [30].

### 4.3. Effects of No-Tillage on GHG Emissions from Intercropping Systems

More sustainable cultivation systems, such as no-tillage, are increasingly used worldwide because of their environmental advantages. Studies on traditional monoculture have shown that no-tillage management reduced soil disturbance, reduced the loss of soil organic carbon, and facilitated the utilization of activated carbon by soil microorganisms. Therefore, soil $CO_2$ emissions were suppressed [42]. A meta-analysis showed that no-tillage decreases GHG emissions with no crop yield tradeoff at the global scale [43]. To evaluate the crop yield and carbon mitigation of intercropping systems, field experiments of maize–pea intercropping with no-tillage and conventional tillage were carried out by Yang et al. [38]. It was found that, over the growing season of maize and pea, no-tillage reduced carbon emissions by an average of 10% compared to conventional tillage, and no-tillage reduced carbon emissions from the maize strips. The effect of no-tillage on reducing GHG emissions was also demonstrated in other cereal–legume intercropping systems [44–46]. However, some scholars believe that the biophysical soil conditions of different no-tillage soil determined the GHG emissions. A reliable evaluation can only be made based on the local situation [47]. Therefore, more research is required in the future.

### 4.4. Effects of Crop Residue Retention on GHG Emissions in Intercropping Systems

Crop residue resources are ubiquitous in crop production industries and play a decisive role in conservation and regenerative agriculture for sustainable development and food security. Crop residue retention in the field is currently the economic utilization of the residue. It is also essential to improving farmland soil health and carbon sink capacity. Crop residue retention and the introduction of legume crops will inevitably increase soil biodiversity, microbial activities, soil carbon, and nitrogen processes, resulting in changes in soil GHG emissions and nitrogen availability [9,48]. Long-term field experiments showed that crop residue retention or the intercropping of legume crops could significantly improve biophysical soil properties and increase the soil organic matter content and crop yields [49,50]. The crop residue nitrogen content and C/N influence soil GHG emissions [51,52]. Therefore, crop residue retention is usually combined with applying nitrogen fertilizer to regulate soil C/N, reducing GHG emissions and improving nitrogen use efficiency [53].

Crop residue retention promotes soil respiration and increases $CO_2$ emissions [53]. The amount of residue retained also significantly affects soil $CO_2$ emissions [54]. However, the mechanisms of soil $N_2O$ emission are complex, there are different views on the effect of residue retention on upland $N_2O$ emission, and the influencing mechanisms need to be clarified [55]. Studies have shown that residue retention can promote soil $N_2O$ emissions, but other studies have concluded that crop residue retention can reduce or not affect soil $N_2O$ emissions [45,56]. In addition, the effects of residue retention on soil $N_2O$ emissions

were significantly correlated with the residue carbon input and crop residue ground cover depth [57,58]. Therefore, the relationship between the quantity and quality of crop residue and biophysical soil properties should be considered when studying soil $N_2O$ emissions. There may be an interaction between residue retention and intercropping regarding soil GHG emissions, but there need to be more relevant studies. More field observations are still required to accurately evaluate the contribution of residue retention to GHG emissions and analyze its influencing factors.

## 5. Problems and Prospects

Recent studies have shown that countries with higher effective crop populations and species diversity have greater temporal stability in terms of total national agricultural output [59]. Agriculture diversity has been proposed to achieve win–win choices among ecosystem services in agriculture production [60,61]. There have been many studies on the effects of cereal–legume intercropping on crop yields. With the intensification of global climate change, more attention has been given to GHG emissions from intercropping systems [9,62–64]. However, the three following aspects still need to be studied:

(1) Compound effect of multi-technology. Through a field experiment conducted for 12 consecutive years (2006–2017), Chai et al. [65] found that multi-technology (e.g., no-tillage or less tillage, crop residue mulching, crop rotation, Nitrogen fertilizer timing, and other agronomic measures, akin to "conservation agriculture") in the intercropping system could significantly increase the yield and optimize ecological benefits. Compared with traditional monoculture, the integrated intercropping system increased the annual profit by 15.6–49.9% and the net agricultural income by 39.2% and reduced the environmental footprint by 17.3%. However, most studies on the characteristics of soil GHG emissions from intercropping are single-factor experiments, and the research on the synergistic effects of cereal–legume intercropping and agronomic conservation practices on GHG emissions is still minimal, which is not conducive to the evaluation and understanding of the multi-technology synergistic impacts of conservation and regenerative cropping systems. Crop residue retention effectively increases the soil organic matter content, especially in areas where soil fertility is declining (e.g., Northeast China). However, both experimental and model studies showed that residue retention could stimulate the emission of $N_2O$ in dry farmland and become the primary "leakage" factor in the carbon sequestration process [47,55,66]. However, the mechanism causing the increase in $N_2O$ emissions needs to be clarified [55]. Whether multi-technology could achieve the synergy of crop productivity and environmental benefits has yet to be well determined.

(2) Indirect GHG emissions. Agricultural carbon footprint theory can systematically evaluate the direct and indirect carbon emissions caused by human factors in agricultural production, which is essential for sustainable agriculture [67]. Life cycle assessment (LCA)-based carbon footprint assessment proved to be an efficient method for quantifying greenhouse gas emissions [68]. Only some studies have considered carbon sequestration in cropping ecosystems and indirect GHG emissions from inputs such as fertilizers and pesticides in cereal and legume production [29,69]. Although some studies have shown that maize intercropping has a higher production efficiency and less of an environmental footprint in arid northwest China [70,71], most intercropping systems still face a trade-off between improving productivity and reducing the carbon footprint [72]. This aspect needs to be more conducive to a comprehensive evaluation of the contribution of various production and market activities in crop production to GHG emissions and a thorough evaluation of the future development potential of the cereal–legume intercropping system as economically and environmentally sustainable.

(3) Non-growing season GHG emissions. According to studies, GHG emissions from farmland are mainly concentrated in the growing season, but the GHG emissions from cropland in the non-growing season must be addressed [33]. Our previous study showed that the $N_2O$ emission contribution of farmland soil in the "black soil" region of northeast China [73] in the non-growing season could reach 15–46% [72]. In years with high snowfall,

$N_2O$ emissions during the spring thaw accounted for 54–76% of the annual total, leading the yearly emissions [32]. However, understanding the drivers and processes underlying $N_2O$ emissions related to freeze–thaw events in residue retention no-till farming systems still needs to be vastly improved. Moreover, there are few studies on GHG emissions under these conditions, which may lead to a miscalculation of the GHG emission reduction effect of managing the entire intercropping system, especially in seasonal freeze–thaw zones.

Sustainable food production is a fundamental challenge in a global warming scenario. An intercropping approach to sustainable food production is crucial to overcoming this global issue. Intercropping has been practiced worldwide in traditional and sustainable agriculture to feed the growing population. However, compared with conventional cereal planting systems, the research on the GHG emissions of a cereal–legume intercropping system in non-growing seasons needs to be more extensive. In situ observations of GHG emissions from intercropping systems in different regions and research on the synergistic mechanism of soil carbon sequestration and GHG emission reduction [74,75] should be strengthened in the future to improve the cereal–legume intercropping technology system according to local conditions and further explore the GHG emission reduction potential of cereal crop intercropping systems (Figure 3).

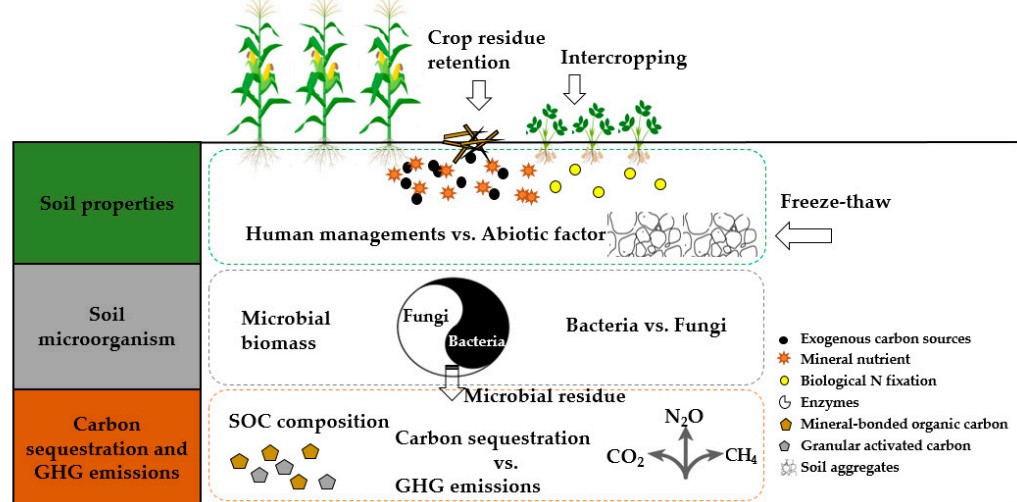

**Figure 3.** Schematic diagram of crop residue retention, intercropping, and freeze–thaw effects on soil organic carbon sequestration and GHG$_S$ emissions.

**Supplementary Materials:** The following supporting information can be downloaded at: https://www.mdpi.com/article/10.3390/agronomy13041085/s1, File S1: 839 literatures.

**Author Contributions:** Conceptualization, F.Y. and Y.W. (Yongjun Wang); methodology, F.Y. and Y.W. (Yang Wu); software, Y.C. and X.L.; validation, F.Y., Y.W. (Yang Wu), X.L., Y.C., Y.L., W.W., W.X., Z.L., J.L. and Y.W. (Yongjun Wang); formal analysis, Y.L., W.W., W.X., Z.L. and J.L.; investigation, X.L. and Z.L.; resources, Y.W. (Yang Wu) and X.L.; data curation, Y.L. and J.L.; writing—original draft preparation, F.Y.; writing—review and editing, Y.W. (Yang Wu) and Y.W. (Yongjun Wang); visualization, Y.C. and Y.W. (Yang Wu); supervision, Y.W. (Yongjun Wang); project administration, Y.W. (Yongjun Wang); funding acquisition, F.Y. All authors have read and agreed to the published version of the manuscript.

**Funding:** This research was funded by the National Natural Science Foundation of China (32001189), the Youth Growth Technology Project of Jilin Province (20210508017RQ), and the Project of Science and Technology Development Plan of Jilin Province (20220404007NC).

**Data Availability Statement:** Data are available on request from the first author.

**Acknowledgments:** We thank the three anonymous reviewers and the editor for their helpful comments and suggestions that have improved this manuscript. We also thank Allen David McHugh, University of Southern Queensland Australia, for checking the manuscript and for the English improvement.

**Conflicts of Interest:** The authors declare no conflict of interest.

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
