# Peer review of "Research Progress and Development Trends of Greenhouse Gas Emissions from Cereal–Legume Intercropping Systems"

_agronomy, doi:10.3390/agronomy13041085_

Round 1
Reviewer 1 Report
Generally, this is a very interesting manuscript to investigate the research progress and development trends on GHG emissions from cereal-legume intercropping systems. I would like to recommend minor revision, and below are detailed comments and suggestions.
1. Replace the word "review" in line 1 with “Perspective”.
2. The the famliy name “Lyu” in line 4 should be consistent with other “Lv”.
3. Replace “The present review” in line 23 with “This work”.
4. Line 41 needs to be indent by 2 characters.
5. In line 236 "...... N availability "should be followed by references.
6. Supplement the shortcomings of the research on soil N2O emission after straw returning to the field in part 5 Problems and prospects (2), such as: the problem of trade-off of carbon sequestration and N2O emission reduction.
7. Check the format of the references 1 and 4.
8. The“-”format between the page numbers of the reference should be uniform.
9. .It is better if the manuscript needs careful editing by someone with in technical English.
10. Could you add a schematic diagram in the conclusion or last part to show the progress and the mechanism of GHG emissions from cereal-legume intercropping, which is different from the monoculture?
11. Please add a table to show the data you collected from literatures.
12. The left top corner of fig 2 is not clear, please improve it.
Reviewer 2 Report
The topic of this review is interesting dealing with a research field acquiring increasing interest in the last years. However, the submitted version should be amended before the acceptance.
The cited literature mainly regards Chinese publications. I think that there are also interesting investigations carried out in others countries. Moreover, the presentation of the results provides uncertain conclusions.
To improve the scientific value I suggest adding further details such as:
Have been described differences of gas emission in relations to geographic areas (mountain, hilly, etc.), soil type, and climatic conditions?
Have been identified methodologies more appropriate to quantify gas emissions?
How the attribution of results was done for papers done by scientists working in different Institute or country?
I suggest replacing in the whole manuscript the symbol N (free nitrogen atom does not exist) with the word nitrogen.
The authors frequently used the words negligible, significant and so on refereed to the comparison of quantitative data. They are very non-specific definitions so does not allow understanding the actual similarity or dissimilarity of presented results.
Line 25: cereal-legume intercropping is a useless keyword being already present in the title.
Lines 52-53: “Many studies have showed….” But only 1 reference is reported. Please add other references.
Line 71: Why cereals were not considered?
Line 75: the criteria adopted for the manual screening of papers should be detailed
Line 238 replace “many studies believe” with many studies shown or demonstrate
The topic of this review is interesting dealing with a research field acquiring increasing interest in the last years. However, the submitted version should be amended before the acceptance.
The cited literature mainly regards Chinese publications. I think that there are also interesting investigations carried out in others countries. Moreover, the presentation of the results provides uncertain conclusions.
To improve the scientific value I suggest adding further details such as:
Have been described differences of gas emission in relations to geographic areas (mountain, hilly, etc.), soil type, and climatic conditions?
Have been identified methodologies more appropriate to quantify gas emissions?
How the attribution of results was done for papers done by scientists working in different Institute or country?
I suggest replacing in the whole manuscript the symbol N (free nitrogen atom does not exist) with the word nitrogen.
The authors frequently used the words negligible, significant and so on refereed to the comparison of quantitative data. They are very non-specific definitions so does not allow understanding the actual similarity or dissimilarity of presented results.
Line 25: cereal-legume intercropping is a useless keyword being already present in the title.
Lines 52-53: “Many studies have showed….” But only 1 reference is reported. Please add other references.
Line 71: Why cereals were not considered?
Line 75: the criteria adopted for the manual screening of papers should be detailed
Line 238 replace “many studies believe” with many studies shown or demonstrate
Reviewer 3 Report
This review focuses on maize legume intercropping, and the associated greenhouse gas emissions. The review is framed from the perspective of increasing yields, while reducing carbon emissions.
The stated objectives of the study are 1) to analyze the development trend of cereal-legume intercropping through a bibliometric method, 2) to summarize the emission characteristics and mechanism of soil N2O and CO2 and elaborate the main factors affecting soil GHG emissions from intercropping systems, 3) to put forward the problems in the study of greenhouse gas in current cereal-legume intercropping systems.
The review needs work to properly summarise the data with appropriate insight from authors. Currently it is a jumbled mash of general research results with very limited insights into intercropping. Eg section 4.2 doesn’t really tell the reader anything. Much of the paper isn’t really specific to intercropping eg section 4.4 which is a summary of the benefits of straw return on soil carbon. The review has also missed significant chunks of relevant literature. There are several reviews and research papers available on maize-legume intercropping; very few have been referenced here despite the long reference list… Such as Maitra, Shankar et al. (2020) Ngwira, Aune et al. (2012) Ananthi, Amanullah et al. (2017) or Yilmaz, Atak et al. (2008) or even Xu, Li et al. (2020). Xu in particular is very similar to this paper conducting “a meta analysis to assess land and fertilizer N use efficiency in intercropping of maize and soybean as compared to sole crops, based on 47 studies reported in English and 43 studies reported in Chinese”. The review by Xu provides a more coherent analysis of the literature with appropriate conclusions drawn.
Currently this review needs a significant revision focusing on a more coherent analysis of current literature with improved focus on intercropping and GHG emissions, rather than GHG emissions generally.
Keywords could be improved. “different management measures” is not a good keyword.
Authors refer to cereal legume intercropping. Eg last sentence of Abstract. Should be changed to maize – legume for consistency.
L37-39: Not clear.
L32-34: Why only reference China, if this is a global problem??
L49: This sentence isn’t true. Soils used for sole cropping are the same soils used for intercropping. It’s possible that intercropping will have an effect on the soil physical and chemical properties!
L65: Point 3 isn’t clear.
L97: Caption for Figure 2 is not adequately descriptive. What do the lines mean? What do the circles mean? Etc.
L101: we got?
L104: top five not top5.
L105: Is it appropriate to assume that “tillage” and “weed” are linked for tillage practices on weed control?? Is it possible that weed control or suppression is not always linked to tillage.
L124: No reference given for this statement!
L127: Soybean intercropping?
L134: “most studies”? No references given at all.
L136: “mainly in the maize strip”. Only one reference, but you give the impression that this is a trend across studies….
L141: Only in upland field??
L147: Sentence needs to be rewritten to make sense.
L164: What defines “reasonable field configuration”?
L172: Once again the authors refer to most studies without referencing anything!!
L187: Authors refer to 1.5 degree warming in the introduction.
L189: Sentence needs re-write.
L195: Sentence???
L207-212: Do these results refer to intercropping systems??
L222:????
L252: See previous comments on references authors havent used. Xu found 47 studies reported in English and 43 studies reported in Chinese on maize soybean alone!
L267: This whole section is questionable in value.
L293: What does this have to do with intercropping?
Ananthi, T., M. M. Amanullah and A. R. M. S. Al-Tawaha (2017). "A review on maize-legume intercropping for enhancing the productivity and soil fertility for sustainable agriculture in India." Advances in environmental biology 11(5): 49-64.
Maitra, S., T. Shankar and P. Banerjee (2020). "Potential and advantages of maize-legume intercropping system." Maize-Production and Use: 1-14.
Ngwira, A. R., J. B. Aune and S. Mkwinda (2012). "On-farm evaluation of yield and economic benefit of short term maize legume intercropping systems under conservation agriculture in Malawi." Field crops research 132: 149-157.
Xu, Z., C. Li, C. Zhang, Y. Yu, W. van der Werf and F. Zhang (2020). "Intercropping maize and soybean increases efficiency of land and fertilizer nitrogen use; A meta-analysis." Field Crops Research 246: 107661.
Yilmaz, Åž., M. Atak and M. Erayman (2008). "Identification of advantages of maize-legume intercropping over solitary cropping through competition indices in the East Mediterranean Region." Turkish Journal of Agriculture and Forestry 32(2): 111-119.
